# Extending the Coverage Area of Regional Ionosphere Maps Using a Support Vector Machine Algorithm

Mingyu Kim[1] and Jeongrae Kim[1]

[1]School of Aerospace and Mechanical Engineering, Korea Aerospace University, Goyang-si, 10540, Korea

**Correspondence to:** Jeongrae Kim (jrkim@kau.ac.kr)

**Abstract.** The coverage of regional ionosphere maps is determined by the distribution of ground monitoring stations, e.g. GNSS receivers. Since ionospheric delay has a high spatial correlation, ionosphere map coverage can be extended using spatial extrapolation methods. This paper proposes a support vector machine (SVM) to extrapolate the ionosphere map data with solar and geomagnetic parameters. One year of IGS ionospheric delay map data over South Korea is used to train the SVM algorithm. Subsequently, one month of ionospheric delay data outside the input data region is estimated. In addition to solar and geomagnetic environmental parameters, the ionospheric delay data from the inner data region are used to estimate the ionospheric delay data for the outside region. The accuracy evaluation is performed at three levels of range – 5˚, 10˚, and 15˚ outside the inner data regions. The extrapolation errors are 0.33 TECU for the 5˚ region and 1.95 TECU for the 15˚ region. These values are substantially lower than the GPS Klobuchar model error values. Comparison with another machine learning extrapolation method, the neural network, shows a substantial improvement of up to 26.7%.

## 1  Introduction

Ionospheric delay is one of the main error sources for single-frequency global navigation satellite system (GNSS) receivers. Ionosphere models or ionosphere maps can be used to correct for ionospheric delay. For real time applications, a regional ionosphere map using regional GNSS monitoring stations can be used to provide highly accurate corrections. The regional ionosphere map coverage is determined by the distribution of GNSS ground monitoring stations. Since ionospheric delay has a high spatial correlation, ionosphere map coverage may be extended by using spatial extrapolation methods. In addition to the spatial correlations, time variables such as observation hour and day number, and solar/geomagnetic indices can serve as input parameters for the extrapolation.

A series of research studies have been conducted on the temporal extrapolation (prediction) of regional ionosphere maps using past observations. With respect to using machine learning algorithms, Kumluca et al. (1999) applied the neural network (NN) method to forecast ionospheric critical plasma frequencies, $f_oF_2$. McKinnell and Friedrich (2007) used an NN to predict the lower ionosphere in the aurora zone. Okoh et al. (2016) developed a regional VTEC model for Nigeria based on observational data from 12 stations and tested temporal and spatial extrapolation performance. Unlike previous studies, the extrapolation performance was improved by adding the International Reference Ionosphere (IRI) as an input. Razin and Voosoghi (2016) applied a wavelet NN with particle swarm optimization to predict the TEC over Iran. Huang and Yuan (2014) used time and temporal variation of the TEC values as radial-basis function (RBF) network inputs to temporal extrapolation. A support vector machine (SVM) model has been used to predict the ionospheric $f_0F2$ above Chinese stations (Ban et al. 2011, Chen et al. 2010). Akhoondzadeh (2013) used a SVM to predict the TEC and to detect seismo-ionospheric anomalous variations.

On the other hand, research on the spatial extrapolation of the ionosphere map is sparse. Wielgosz et al. (2003) used kriging and multiquadric method to produce instantaneous TEC maps near the Ohio CORS stations in near-real time. Kim and Kim (2014) applied a biharmonic spline method to extend a small ionospheric correction coverage area. Ionospheric delay observations were used as the input parameters, and the ionospheric delay outside the coverage area was extrapolated. Leandro and Santos (2006) used geographical information as inputs of a NN model for spatial extrapolation of TEC over Brazil. For spatial extrapolation, Jayapal and Zain

(2016) used a NN with time and solar/geomagnetic indices. In addition to these environmental parameters, Kim and Kim (2016) used the ionospheric delay of the inner area to improve the performance of spatial extrapolation.

In addition to the NN method, a SVM algorithm can be considered for spatial extrapolation. An SVM finds a solution to the convex quadratic programming problem in training to optimize the margin so that it can be both optimal and unique. On the other hand, an NN finds the weight between each layer through the gradient descent method, and the solution has a possibility to fall into the local minima in this process. An NN is based on empirical risk minimization (ERM), which is a method of minimizing learning errors during the learning process. On the other hand, an SVM is based on structural risk minimization (SRM), so it has excellent generalization performance (Gunn, 1998). SVMs have been widely used as predictive models in various fields. Huang et al. (2015) successfully performed stock market movement predictions using an SVM. Mohandes et al. (2014) performed wind speed predictions using an SVM and compared the performance against the NN method. The results showed that the SVM achieved superior prediction performance.

This paper proposes an SVM algorithm to extend ionosphere map coverage by applying temporal/environmental parameters and ionospheric observations. The IGS ionosphere map is used as a reference map, and the extrapolation accuracy of the SVM is evaluated by comparing it to the IGS map data. The extrapolation accuracies are compared with the GPS Klobuchar model and the NN model.

## 2   Parameter Modeling

Three types of input parameters are used for the extrapolation of a regional ionosphere map – temporal parameters, environmental parameters, and ionospheric delay observations. An extrapolated ionospheric delay, $ID_{ext}$, may be represented as a function of these three parameters.

$$ID_{ext} = f\left(x_t \quad x_e \quad x_{obs}\right) \tag{1}$$

where $x_t$ and $x_e$ are the time and the environmental parameters, respectively. $x_{obs}$ is ionospheric delay observations in the inner area. The inner area is defined as a geographical area where ionospheric delay information or observations are available. The outer area is defined as a geographical area where ionospheric delay will be estimated.

The ionospheric variation is correlated with the diurnal and seasonal time variation, and the ionospheric delay above the locations involved in the study reaches its maximum around 14 hours local time (LT) and its minimum around 2 LT (Wu et al. 2012). Also, the daily mean ionospheric delay is higher in spring and autumn, and lower in summer and winter (Wu et al. 2012, Mansoori et al. 2015). In order to adopt these correlations, time parameters are included in the extrapolation model. The diurnal variation is represented by an hour number (0 ~ 23 LT), and the seasonal variation is represented by a day number (0 ~ 365). To represent the repeatability of these variations, the time parameters are modeled as sinusoidal functions.

$$x_t = \left[S_D \quad C_D \quad S_H \quad C_H\right] \tag{2}$$

where $S_D$ and $C_D$ are the sine and cosine, respectively, of the day number, and $S_H$ and $C_H$ are the sine and cosine, respectively, of the hour numbers. The periods used for the sinusoidal functions are set to 24 hours and 365.25 days for the diurnal and seasonal parameters, respectively. The ionosphere activity is also highly correlated with solar and geomagnetic activity. Three parameters are selected to reflect the space environment – the F10.7 index, geomagnetic index Kp, and sunspot number (SSN).

$$x_e = \left[F10.7 \quad Kp \quad SSN\right] \tag{3}$$

Although SSN has a similarity with F10.7 in representing solar activity, use of both parameters yielded slightly better estimation accuracy than use of single parameter. Therefore, both F10.7 and SSN are adopted for the environmental parameters. Experiments on selection of optimal solar activity indices will be discussed in Section 5.

Disturbance storm time (Dst) may replace Kp for ionosphere storm detection but it was not selected. Dst response performance depends on ionosphere storm driver. Dst is efficient for coronal mass ejection (CME)-driven storms, but it is less effective for corotating interaction region (CIR) / coronal hole high speed streams (CH HSS) driven storms (Borovsky and Denton 2006, Denton et al., 2006). After series of numerical experiments on selecting Dst or Kp, Kp has been selected for the parameter because of its better estimation performance. The numerical experiments will be discussed in Section 4.

Past inner-area ionospheric delays are used to train the machine learning algorithms, and current inner-area delays are used for the extrapolation. The observation data set for the N observation points is derived as follows.

$$x_{obs} = \begin{bmatrix} ID_{obs}^1 & ID_{obs}^2 & \cdots & ID_{obs}^N \end{bmatrix} \tag{4}$$

The proposed algorithm is using fixed locations both for input and output, and it does not require a spatial structure. Other researchers' works on ionosphere prediction used raw GPS TEC measurements at varying IPP (Ionospheric Piercing Point) and the measurement locations should be registered in the input. Our algorithm uses a grid-based ionosphere map with fixed grid points, and their location information is not required as the model inputs.

In the event of high temporal or geographical decorrelation due to geomagnetic storm, two inputs are affected: the solar/geomagnetic parameters and the ionosphere input data in inner region. Because of observation latency, the real-time solar/geomagnetic parameters may not be available in real time. However the ionosphere input data may be available in real time from GPS observations, and this fact makes for the estimation algorithm to respond to the geomagnetic storm in real time.

## 3   Extrapolation methods

### 3.1   Support vector machine (SVM)

The SVM method is a machine learning theory proposed by Vapnik in 1995. It uses an algorithm to find a hyperplane that maximizes the margin (Gunn, 1998). It is used in data classification and regression problems, and SVMs used in regression are referred to as support vector regression (SVR). An SVM sets the regression function, $f(x_{svm})$, such that target $y_{svm}$ is in the following range.

$$f(x_{svm}) = \hat{y}_{svm} = w^T x_{svm} + b \tag{5}$$

$$f(x_{svm}) - \varepsilon \leq y_{svm} \leq f(x_{svm}) + \varepsilon, \ \varepsilon > 0 \tag{6}$$

where $x_{svm}$ is the input that contains $\begin{bmatrix} x_t & x_e & x_{obs} \end{bmatrix}$, and $w^T$ is the transposed weighting matrix. $y_{svm}$ is the target that represents the true ionospheric delay in the extrapolation region. $x\varepsilon$ is the allowable error level for $y_{svm}$. In many practical cases, $y_{svm}$ is not in the range of $(f(x_{svm}) - \varepsilon, f(x_{svm}) + \varepsilon)$, and $y_{svm}$ is frequently adjusted to the range of $(f(x_{svm}) - \xi, f(x_{svm}) + \xi)$, where $\xi$ is a slack variable. The optimal regression function is determined when the total magnitude of the slack variable, $\sum_i \xi_i$ is minimized. Also, the distance between $f(x_{svm})$ and the support vector should be maximized. The distance between the SVM and $f(x_{svm})$ is called the margin, and the margin may also be minimized. Therefore, the optimal regression function minimizes $\|w\|$ and $\xi$ to achieve the maximum margin (Gunn, 1998).

$$\min \frac{\|w\|^2}{2} + C \sum_{i=1}^n \left( \xi_i^- + \xi_i^+ \right) \tag{7}$$

*Subject to* $y_{svm} - f(x_{i,svm}) - \xi_i \leq \varepsilon, \quad if \quad y_{svm} - f(x_{i,svm}) \geq \varepsilon$

$$y_{svm} - f(x_{i,svm}) + \xi_i \geq -\varepsilon, \quad if \quad y_{svm} - f(x_{i,svm}) \leq -\varepsilon \tag{8}$$

In equation 7, the superscript – denotes a lower boundary and + denotes an upper boundary. The slack variable disappears while expanding equations. $C$ is the penalty set by users. As the $C$ value approaches zero, the weight for the slack variable decreases and the relative weight for $\|w\|^2$ increases. Therefore, the regression function that maximizes the margin can be calculated. This implies that the regression function differs from $y_{svm}$. As $C$ increases, the weight for the slack variable sum increases rather than maximizing the margin magnitude. Therefore, a regression function is calculated in a form similar to $y_{svm}$. Eq. (7) can be modified using a dual problem, as follows.

$$\arg\min_{\beta} \frac{1}{2} \beta^T K\left(x_{i,SVM}, \quad x_{j,SVM}\right) \beta - f^T \beta, \quad f = -y_{SVM} + \varepsilon \tag{9}$$

Where $\beta$ is $\alpha^- - \alpha^+$ and $\alpha$ is Lagrange multiplier. $K$ is a kernel function that maps input data $x_{svm}$ to a higher dimension. Kernel functions have several functions, including linear and polynomial functions. The most commonly used functions are Gaussian kernel functions (Cristianini, 2001).

$$K\left(x_{svm}, y_{svm}\right) = \exp\left(-\frac{\|x_{svm} - y_{svm}\|^2}{2\sigma^2}\right) \tag{10}$$

After mapping $x_{svm}$ to feature space, one can determine the optimal $\beta$ by using quadratic programming (QP). The optimal regression function can be computed by using the following equation (Gunn, 1998).

$$f\left(x_{svm}\right) = w^T x + b = \sum_{i=1}^{N} \beta^T K\left(x_{i,SVM}, \quad x_{j,SVM}\right) + \frac{1}{n} \sum_{i=1}^{N} \sum_{j=1}^{N} \left\{ y_{i,SVM} - \beta_j^* K\left(x_{i,SVM}, \quad x_{j,SVM}\right) \right\} \tag{11}$$

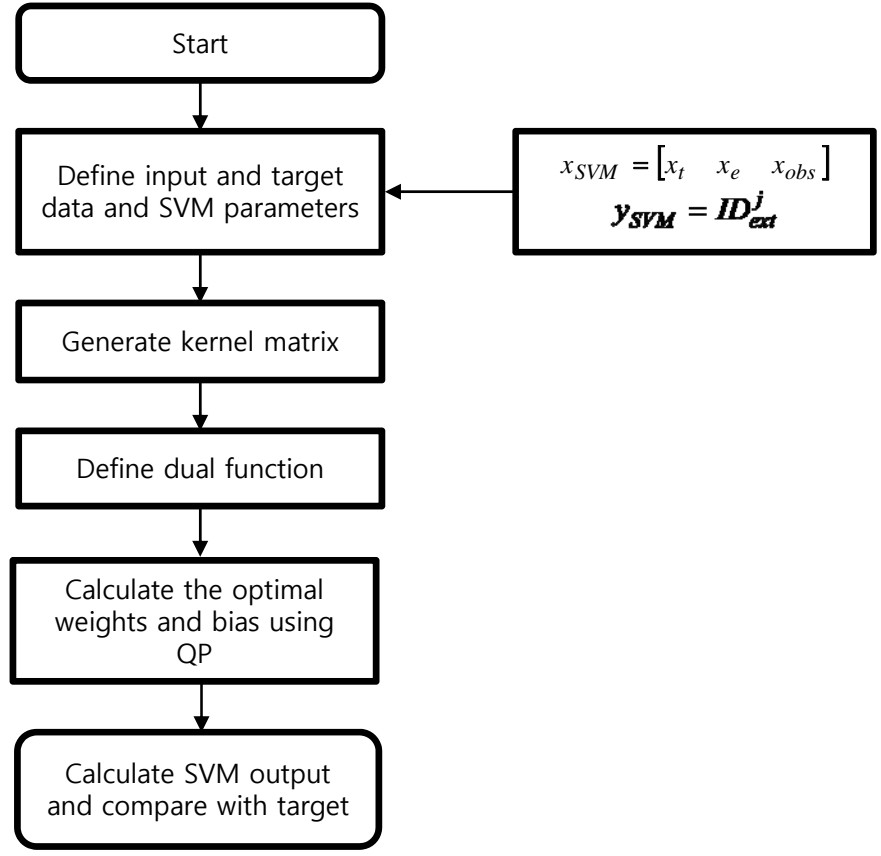

**Figure 1: Flow chart of the SVM training process**

The flow chart of the SVM training process is shown in Figure 1. The input variables consist of temporal and environmental parameters and ionospheric delays in the observation region, and these inputs are identical for each extrapolation point. Targets include the true ionospheric delay in the j-th extrapolation point. After the input/output of the SVM is defined, a kernel matrix is generated for each input. Then, the training is performed to find the optimal coefficients and bias of the regression function, $f(x_{svm})$. The kernel function is calculated for the epoch of each input so that the size of the matrix becomes $N \times N$, where $N$ is the number of epochs. As the input increases, the computational time and memory usage also increase. Therefore, the elements of the kernel matrix, including the oldest epoch, are deleted, and the kernel functions of the recent epoch are included in the matrix. After defining the kernel function and the boundary of the regression function, the optimal weights and biases are calculated using the interior point method (Ferris and Munson, 2004). When the initial training is completed, the extrapolation and update of the kernel function are repeated.

## 3.2 Neural network (NN)

An NN is a statistical learning model similar to a biological neural network. It consists of neurons or perceptions, and a synapses. Neurons are interconnected with synapses, which store weights. An NN can solve problems such as pattern recognition and regression by calculating the weights from the learning of the neurons (Habarulema et al. 2011).

Several types of NNs exist – e.g. back-propagation neural network (BPNN), recurrent neural network (RNN), and time delay neural network (TDNN). This study implements a BPNN, which is one of the most commonly used NN algorithms. It is a feed-forward, multi-layer perceptron (MLP), supervised learning network

(Jwo et al, 2004). In the hidden layer, activation functions determine whether the values from the previous layer are activated or not. Training is generally performed using gradient descent method.

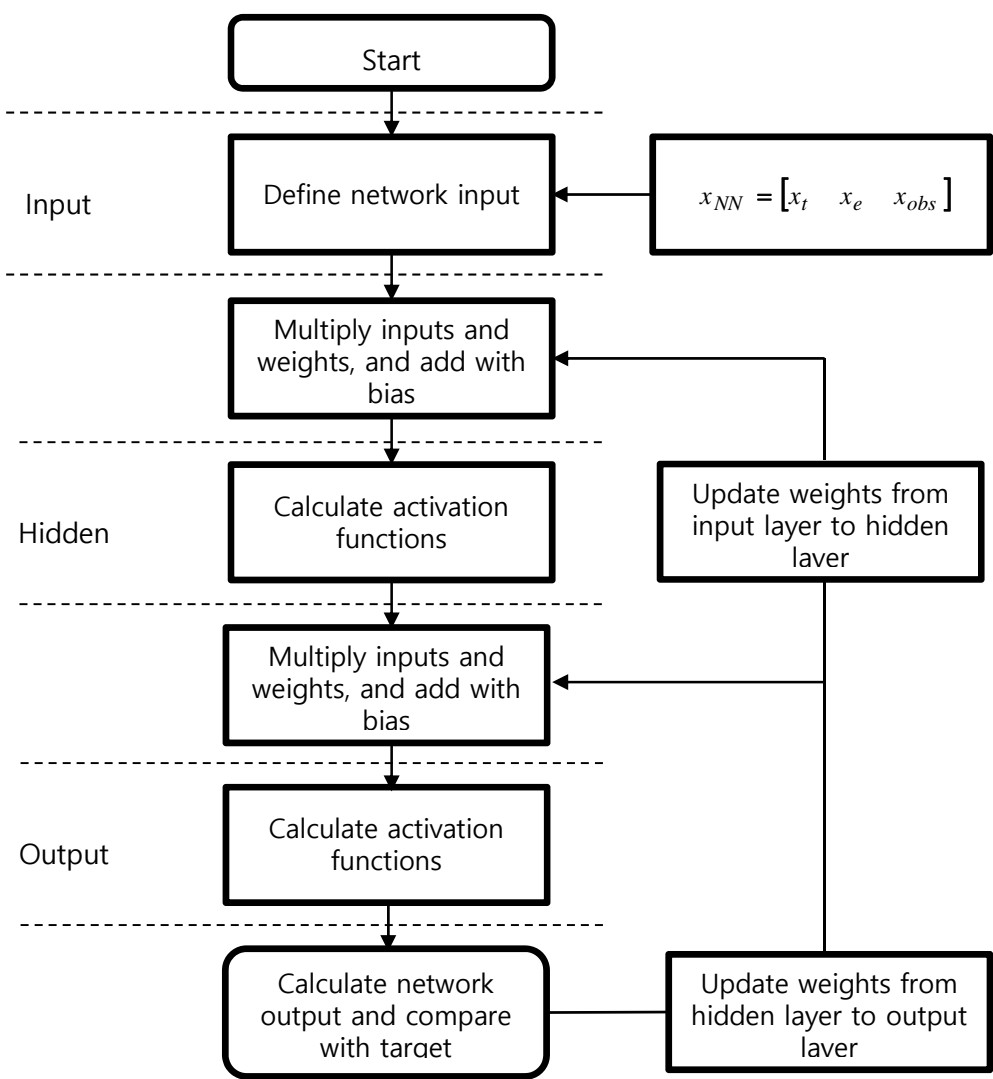

**Figure 2: Flow chart of the neural network training process**

Figure 2 shows a flow chart of the BPNN used for the regional ionosphere map extrapolation. The input layer includes the network inputs, $x_{NN}$, shown in Eqs. (2), (3), and (4). The network inputs and targets are the same as those used in the SVM. An input neuron multiplied by a weight can be computed through the hidden layer towards the output neuron, as follows.

$$\hat{y}_{NN} = f^n(W^{n,n-1}f^{n-1}(W^{n-1,n-2}f^{n-2}(\cdots f^1(W^{1,0}x_{NN} + b^1)\cdots + b^{n-2}) + b^{n-1}) + b^n) \tag{12}$$

where $b$ is the network bias, $n$ represents the n-th layer, and $W^{n,n-1}$ is the weight from $n-1$ to the $n$-th layer. $x_{NN}$ is the network input, which includes the three input parameters for extrapolation, and $\hat{y}_{NN}$ is the network output. $f$ is an activation function. The hyperbolic tangent sigmoid function is implemented, which is the most widely used method. The network is trained using the BPNN algorithm with true ionospheric delays and three

input parameter sets to find the optimal weights and biases.

The network data is generally divided into training, validation, and test sets. The training set is used to calculate and update the weights. The validation set is used to verify the training results. The test set is finally used to calculate the extrapolation error. This paper uses three data sets divided by 70%, 15%, and 15%, respectively. A detailed implementation of the NN can be found in Kim and Kim (2016).

## 4   Data Processing

An IGS global ionosphere map (GIM) is used to acquire reference ionospheric delay data because of its high accuracy and global coverage. Regional ionospheric delay time series are generated with the GIM data, and they are used to train the extrapolation algorithms. The extrapolated ionospheric delays outside the observation area are compared with the GIM data to evaluate the accuracy. The IGS GIM grid size is 2.5° x 5°, but other regional ionosphere maps such as the space-based augmentation system (SBAS) ionosphere corrections have an equal latitude-longitude grid size. Therefore, a 5° x 5° grid size is used for the regional ionosphere map in this research.

The estimation interval is the same as the ionosphere input data interval. In this research, two-hour interval was used because two-hour interval IGS global map is implemented for the inner map. If a shorter interval inner map is used, e.g. 5 min. SBAS map or real-time GPS-derived map, and then the estimation interval becomes shorter. The proposed algorithm is not a time-prediction algorithm, as other preceding researches, and the estimation interval is not an important factor to determine the accuracy.

Figure 3 illustrates the observation and extrapolation grid points. The observation regions (blue) are set with a radius of 2,650 km centered on South Korea, and the extrapolation regions (red) are set with a radius of 4,500 km in order to include the 15° extended grid point from South Korea. Therefore, the latitude of the observation area ranges from 15°N to 55°N, and the longitude ranges from 105°E to 150°E. The accuracy evaluation points are selected to perform the extrapolation. In order to accommodate the directional characteristics of the extrapolation performance, the evaluation point set is selected for each direction (north, south, east, and west). In each direction, three points are selected with different distances from the inner observation region – 5°, 10°, and 15°. All the locations of the extrapolation points are represented in Table 1.

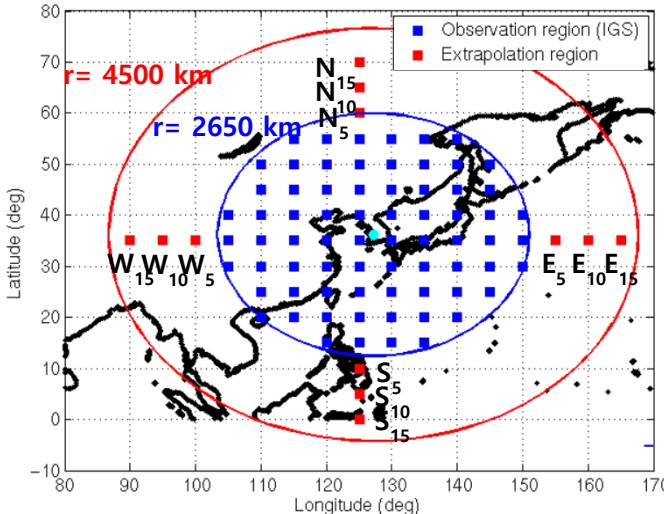

**Figure 3: Observation and extrapolation regions of ionospheric delay grids**

**Table 1: The locations of the extrapolation points**

| Extrapolation point (deg) | North | East | South | West |
|---|---|---|---|---|
| **5** | 60˚N, 125˚E | 35˚N, 155˚E | 10˚N, 125˚E | 35˚N, 100˚E |
| **10** | 65˚N, 125˚E | 35˚N, 160˚E | 5˚N, 125˚E | 35˚N, 95˚E |
| **15** | 70˚N, 125˚E | 35˚N, 165˚E | 0˚N, 125˚E | 35˚N, 90˚E |

In the case with the environmental parameters (i.e. F10.7, Kp, and SSN), real-time data may not exist at the extrapolation epoch due to data latency. In order to simulate this data latency, previous one-epoch (2-hour) values are used instead of the current values during the extrapolation process. This time interval is not large because it is not a temporal prediction method, but a spatial extrapolation method. The influence of the time interval on the estimation performance is much smaller than the ionosphere input data. True environmental parameters are used in the training process, but the previous one-epoch values are used in the extrapolation process. The correlation analysis between the current and previous one-epoch values confirms the correlation. The correlation coefficients between the two adjacent epochs of data for F10.7, Kp, and SSN are 0.930, 0.863, and 0.852, respectively. Since the IGS GIM uses 2-hour intervals, the Kp, which is provided every 3 hours, is interpolated at intervals of 2 hours.

Previous research showed that extrapolation errors have a high correlation with the ionospheric delay magnitude and variation (Kim and Kim, 2014). Therefore, the high ionospheric delay season is more appropriate when evaluating the extrapolation algorithm than the low ionospheric delay season. It means that if the magnitude of the ionospheric delay and variation is small, all the extrapolation values and errors are small. In this case, it is difficult to compare the extrapolation performance for each model. The training period is set to one year from October 1, 2013 to September 30, 2014. In this period, the minimum and maximum ionospheric delays are 5.1 and 112.2 TECU, respectively, as shown in Figure 4. The extrapolation period is set to one month from October 1 to 31, 2014. The region analyzed in this paper is located around the mid-latitude. In this region, the ionospheric spatial gradient is large in the North-South direction. Also, since the southern area is close to the geomagnetic equator, its ionospheric variation is very large.

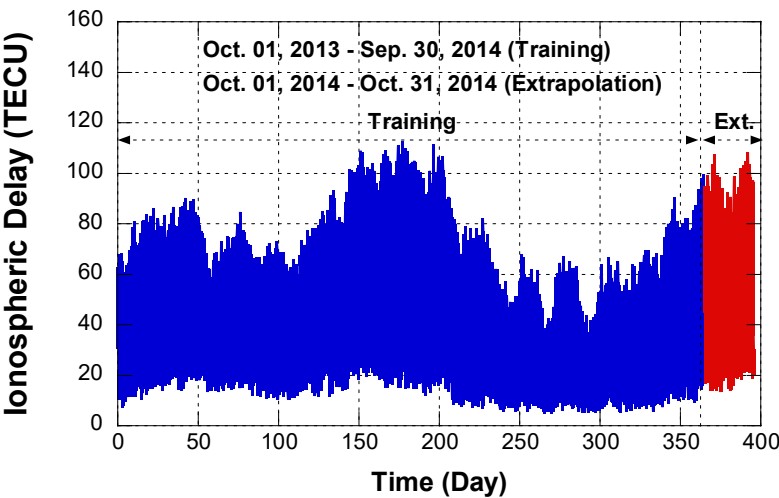

**Figure 4: One year variation of ionospheric delay (October 01, 2013 to October 30, 2014, S15)**

The training and extrapolation performance depend on user parameters. In the case of the NN, extrapolation performance mainly depends on the number of hidden neurons. If the number of hidden neurons is too high, over-fitting may occur, and the calculation time is long. Since there are no criteria for determining the number of hidden neurons, the optimal number of hidden neurons must be found by analyzing the extrapolation error variation due to the number of neurons. The model parameters with the lowest test error are adopted as the optimal values. In Fig. 5 and 6, test errors are computed by the mean RMS extrapolation errors at the 5° extrapolation regions. In case of the NN, the number of hidden neurons was selected as 80 where the error becomes a minimum. In the case of the SVM, the extrapolation result also varies with the model parameters. This paper sets the penalty, $C$, as $10^6$ (Fig. 6), which causes the regression function to almost equal $y$. The Gaussian function, which is widely used in SVMs, is used as a kernel function, and $\sigma$ is set to $10^{-6}$. The values of $\sigma$ and $\varepsilon$ are selected via trial and error to determine the lowest extrapolation error case. They are set to $10^{-6}$ and $10^{-7}$, respectively.

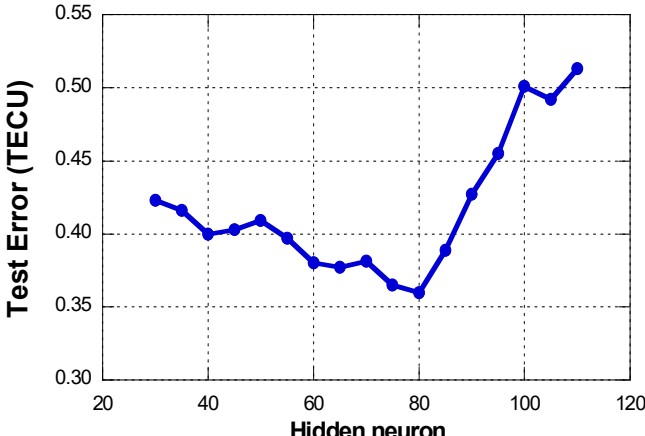

**Figure 5: Test errors of different numbers of hidden neurons by the NN model (5° extrapolation point)**

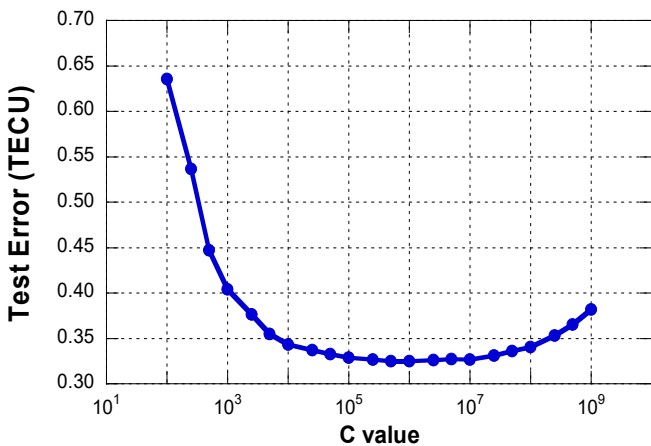

**Figure 6: Test errors of different C values by the SVM model (5° extrapolation point)**

In order to select an ionospheric storm-related input parameter between Kp and Dst, series of experiments had been performed by replacing Kp with Dst. The experiments concluded that Kp is better for our estimation algorithm than Dst is. After replacing Kp with Dst, both the SVM and NN estimation accuracies were

degraded. At 5° extrapolation points, the SVM estimation error was increased from 0.33 TECU (Kp) to 0.44 TECU (Dst) and the NN estimation error was increased from 0.45 TECU to 0.63 TECU. Similar level of error increases were observed at both 10° and 15° points. The NN accuracy degradation with Dst was more significant during high ionospheric disturbance period, when Dst < -25nT (October 9, 19-21, 28). However only one month of data are tested in this research. One month may not be sufficient for evaluating the estimation performance under various ionosphere conditions, e.g. CME- or CIR/CH HSS-related ionospheric disturbances. Comprehensive analysis with a longer data period, e.g. multiple years, can be a further research topic.

## 5    Results

The regional ionosphere map extrapolation is performed using the SVM, and the IGS GIM is used as a truth value. The SVM extrapolation results are compared with the NN and Klobuchar model results. Hourly variations of the extrapolation results are analyzed with one-day data, and then daily variations of the results are analyzed with one-month data.

### 5.1    Single-day extrapolation analysis

The variations of the ionospheric delay and the extrapolation results are analyzed for the data from October 28, 2014, when the daily ionospheric delay magnitude reaches its maximum for the extrapolation period (October 2014).

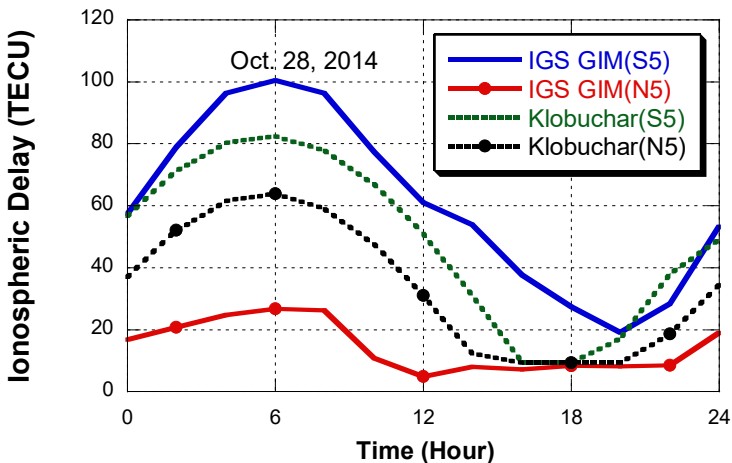

**Figure 7: Ionospheric delays of the IGS GIM and Klobuchar model (south 5˚ and north 5˚ points)**

Figure 7 shows the ionospheric delay variations of the IGS GIM and Klobuchar model on October 28, 2014. Data from two evaluation points, 5˚ north and south are presented. Universal time (UT) is used. The ionospheric delay reaches its maximum at 15:00 LT (6:00 UT) and then decreases. There are large differences between the ionospheric delays at the north and south points because of the ionospheric spatial gradient (Kim et al. 2014). The north-south difference produced by the Klobuchar model is significantly smaller than the IGS GIM.

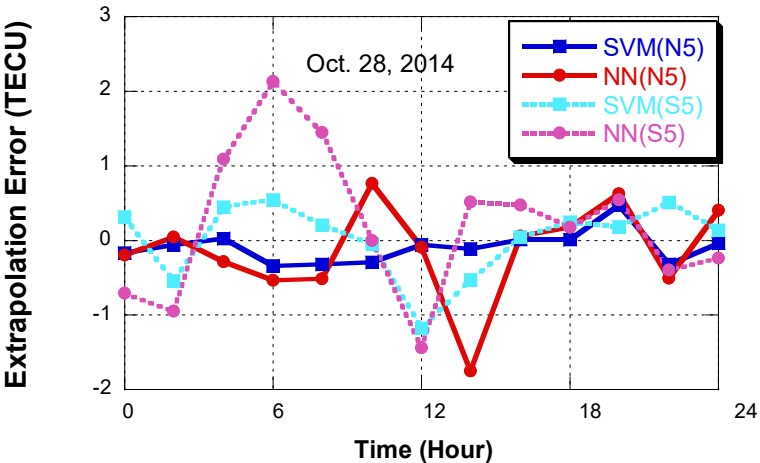

**Figure 8: Extrapolation error variations on October 28, 2014 (north 5° and south 5° points)**

Figure 8 shows the extrapolation results for October, 28, 2014. Two extrapolation points, north 5° (N5) and south 5° (S5), are selected. In the case of N5, the extrapolation RMS errors of the SVM and NN are 0.23 TECU and 0.63 TECU, respectively. The SVM outperforms the NN with a 63.5% error reduction. The NN error increase at 6 UT corresponds the ionosphere maximum at 6 UT in Fig. 7, and the overall NN error variation at S5 follows the ionospheric delay variation. The NN error at N5 and SVM errors at S5/N5 do not follow the ionospheric delay variation.

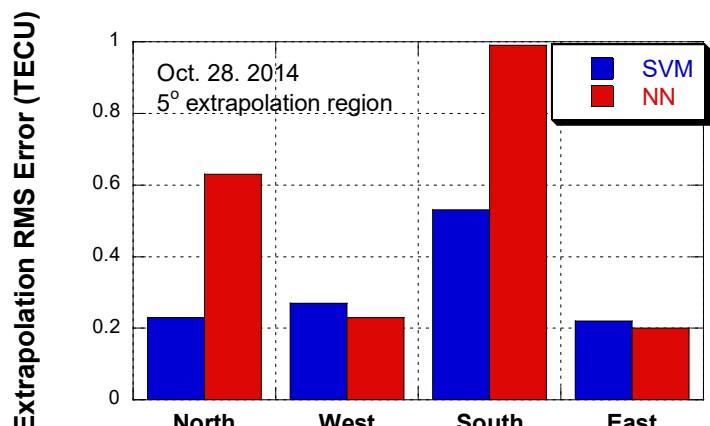

**Figure 9: Extrapolation errors for each direction (5° extrapolation regions)**

Figure 9 compares the RMS errors of four 5° extrapolation points (N5, S5, E5, and W5) on October 28, 2014. The error magnitude is the largest at the south point where the ionospheric delay magnitude is the largest. The SVM shows similar error levels for the north, east, and west points. However, the NN shows larger errors than the SVM even at the north point. This difference in extrapolation accuracy may be explained via the ionospheric spatial gradient. The spatial gradient along the north-south direction is significantly greater than the gradient along the east-west direction (Kim et al. 2014, Vuković and Kos 2016). The large gradient increases the geographical ionospheric delay difference and frequently causes the NN error increase. However, the SVM is

more robust for this large amount of gradient data. In general ionosphere estimation errors increases at low geomagnetic latitude (Song et al. 2018) However, the errors at E5 and W5 are smaller than those at N5 point even though E5 and W5 are located to the south of N5. This is because the input of the model includes the internal ionospheric delay for solving a spatial extrapolation problem. It implies that the ionospheric spatial gradient is the main factor of the extrapolation performances.

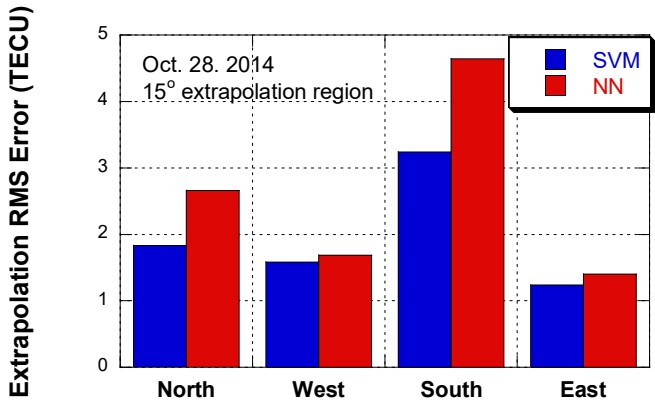

**Figure 10: Extrapolation errors for each direction (10˚ extrapolation regions)**

Figure 10 compares the RMS errors of four 10˚ extrapolation points (N10, S10, E10, and W10) on October 28, 2014. Unlike the 5˚ results in Fig. 7, there is little difference between the two models for the northern area. However, the difference between the two models in the southern region is increased to 0.63 TECU. It means that the extrapolation performance of the SVM and the NN model is larger for the high ionospheric variation region. The extrapolation errors of the East and West region are not significantly different from those in Fig. 9.

**Figure 11: Extrapolation errors for each direction (15˚ extrapolation regions)**

Figure 11 compares the RMS errors of four 15˚ extrapolation points (N15, S15, E15, and W15). The overall error level increases from that of the 5˚ points, but the SVM still outperforms the NN, particularly at the south and north points. The SVM error at the south point is 3.24 TECU, and the error reduction over the NN is 1.40 TECU, or 30.2%. As the extrapolation points become far away from the ionosphere input data points, the extrapolation algorithm efficiency becomes diminished. Therefore, the accuracy difference between SVM and NN has been reduced.

**5.2 One-month extrapolation analysis**

The spatial extrapolations are performed for the one-month period from October 1 to 31, 2014. As with the single-day extrapolation, the one-year data from October 2013 to September 2014 is used for the training process.

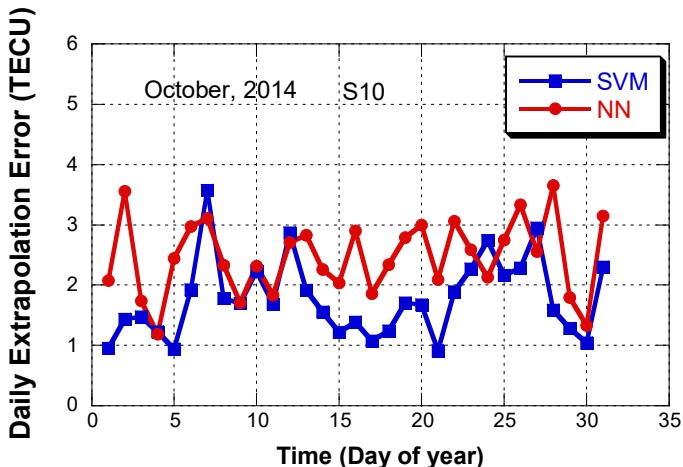

**Figure 12: Daily extrapolation RMS error variations in October 2014 (south 10° point)**

Figure 12 shows the daily extrapolation errors for the south 10° extrapolation point (S10) in October 2014. The one-month means of the daily RMS errors are 1.89 TECU for the SVM and 2.54 TECU for the NN. During the 31 days, the SVM achieved better performance than the NN for 26 days (83.9%). During low ionospheric delay periods, the difference in extrapolation performance between the two methods is not significant (e.g. October 9 and 10). However, during high ionospheric delay periods, the difference becomes significant (e.g. October 28).

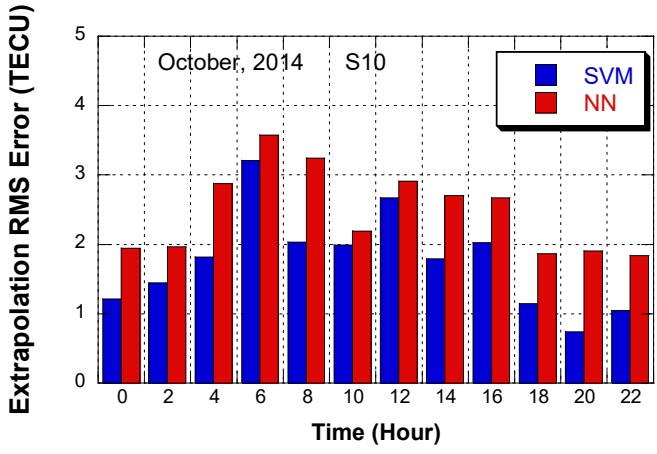

**Figure 13: Extrapolation RMS errors for each two-hour interval on October 2014 (south 10° point)**

In order to analyze the hourly extrapolation performance, the one-month mean of each two-hour time

interval is presented in Figure 13. The time unit is UT. Both the SVM and NN show an increase in extrapolation errors at 06:00 UT. During the high ionospheric variation period, 04:00-08:00 UT, the mean of the SVM error is 0.88 TECU lower than the error of the NN. Even during the low ionospheric variation period, 18:00-22:00 UT, the SVM error is 0.88 TECU lower than the NN. These results prove that the extrapolation performance of the SVM model is better for both large and small ionospheric delays. A correlation analysis with the geomagnetic index, Kp, is performed by computing statistics for each Kp value. (This is not shown as a figure.) Over all Kp values, the SVM outperforms the NN with the same level of improvement. The only exception is Kp= 5 on October 5 12:00 UT, where the NN outperforms the SVM. However, this high Kp happens only one time among 360 epochs, and a generalized conclusion requires a further research.

**Table 2: One-month mean of extrapolation RMS errors using the SVM, NN, and Klobuchar models (unit= TECU)**

| Extrapolation region | 5° | | | 10° | | | 15° | | |
|---|---|---|---|---|---|---|---|---|---|
| | Klob. | SVM | NN | Klob. | SVM | NN | Klob. | SVM | NN |
| North | 14.41 | 0.32 | 0.68 | 13.07 | 1.02 | 1.06 | 12.04 | 1.97 | 1.90 |
| East | 14.63 | 0.17 | 0.20 | 14.57 | 0.51 | 0.71 | 14.47 | 1.00 | 1.13 |
| West | 13.38 | 0.24 | 0.25 | 13.29 | 0.64 | 0.63 | 13.12 | 1.27 | 1.44 |
| South | 25.13 | 0.57 | 0.67 | 24.40 | 1.89 | 2.54 | 26.97 | 3.58 | 3.79 |
| Total | 16.89 | 0.33 | 0.45 | 16.33 | 1.01 | 1.23 | 16.65 | 1.95 | 2.06 |

Table 2 summarizes the extrapolation errors for all evaluation points in October 2014. The one-month mean of the errors from four directions, north, south, east, and west, and three ranges, 5°, 10°, and 15°, are presented. The Klobuchar model of the GPS navigation message (Klob.) is also shown for comparison. In all ranges, even at the 15° points, both the SVM and NN outperform the Klobuchar model. This proves that the extrapolation methods are useful even in large areas. In the east and west points where the ionospheric spatial gradient is small, the accuracy improvement provided by the SVM is not significant because it can be suitable to generalize the ionospheric delay by internal ionospheric delay information. The SVM error is 11.8% smaller than that of the NN in the W15 region. In the south region, the extrapolation error is very large due to the large ionospheric variation, and these results in the largest improvement provided by the SVM. In particular, the S10 region contains the largest error difference at approximately 0.65 TECU. The average error for each region is the largest at the 10° extrapolation region.

The difference may mainly result from the fact that the generalization performance of the SVM model is better than that of the NN for the ionospheric variations. Since ionosphere environment depends on its geomagnetic locations, the proposed extrapolation algorithm performance might be different at other locations. If the estimation region is changed, a new training and optimization process should be performed.

In order to determine optimal parameters between F10.7 and SSN, two more cases are tested; F10.7 only and SSN only. Optimal estimator structure is changing with the selection of input parameters. Before comparing the single parameter (F10.7 only or SNN only) results with the dual parameter (F10.7 and SNN) results, same types of parameter optimizations are performed as Figures 5 and 6 for each single parameter case. SVM C value is selected to 10000 for both cases. The optimal numbers of hidden neurons are selected to 55 for the F10.7 case and 45 for the SSN case.

**Table 3: One-month mean of extrapolation RMS errors with three parameterizations (SVM model, unit=TECU)**

| Extrapolation region | 5° | | | 10° | | | 15° | | |
|---|---|---|---|---|---|---|---|---|---|
| | F10.7 | SSN | SSN+F10.7 | F10.7 | SSN | SSN+F10.7 | F10.7 | SSN | SSN+F10.7 |
| North | 0.31 | 0.31 | 0.32 | 1.11 | 1.14 | 1.02 | 2.05 | 1.98 | 1.97 |
| East | 0.26 | 0.25 | 0.17 | 0.58 | 0.57 | 0.51 | 1.06 | 1.06 | 1.00 |
| West | 0.26 | 0.26 | 0.24 | 0.78 | 0.80 | 0.64 | 1.25 | 1.26 | 1.27 |

| | | | | | | | | | |
|---|---|---|---|---|---|---|---|---|---|
| **South** | 0.66 | 0.67 | 0.57 | 1.95 | 1.93 | 1.89 | 3.62 | 3.65 | 3.58 |
| **Total** | 0.34 | 0.34 | 0.33 | 1.11 | 1.11 | 1.01 | 2.00 | 1.98 | 1.95 |

**Table 4: One-month mean of extrapolation RMS errors with three parameterizations (NN model, unit=TECU)**

| Extrapolation region | 5° | | | 10° | | | 15° | | |
|---|---|---|---|---|---|---|---|---|---|
| | F10.7 | SSN | SSN+F10.7 | F10.7 | SSN | SSN+F10.7 | F10.7 | SSN | SSN+F10.7 |
| **North** | 0.66 | 0.64 | 0.68 | 1.16 | 1.20 | 1.06 | 2.05 | 2.01 | 1.90 |
| **East** | 0.25 | 0.24 | 0.20 | 0.73 | 0.68 | 0.71 | 1.14 | 1.15 | 1.13 |
| **West** | 0.25 | 0.23 | 0.25 | 0.78 | 0.83 | 0.63 | 1.46 | 1.48 | 1.44 |
| **South** | 0.88 | 0.73 | 0.67 | 2.82 | 2.63 | 2.54 | 3.90 | 3.91 | 3.79 |
| **Total** | 0.51 | 0.46 | 0.45 | 1.37 | 1.34 | 1.23 | 2.14 | 2.14 | 2.06 |

The extrapolation RMS errors of the single (F10.7 or SSN) and dual (F10.7+SSN) parameters are presented in Table 2 (SVM) and Table 3(NN). The total mean errors of the single parameter cases are greater than the dual parameter case at all extrapolation points for both estimation models. Increase of the NN errors with the single parameters at North and South points are significant. Effect of F10.7 and SSN may be complementary to each other during geomagnetic storm days (October 19-22). In this period, the estimation error reduction by the dual parameters are 26% for SVM model and 22% for NN model.

## 6   Conclusions

The coverage area of a regional ionosphere map is determined by the distribution of GNSS ground stations. This paper proposes a spatial extrapolation algorithm to extend the ionosphere map coverage using an SVM. One year of IGS GIM ionospheric delay data over South Korea and environmental parameters are used as input data sets to train the SVM algorithm. From the training results, one month of ionospheric delay data outside the input data region is estimated. In addition to solar and geomagnetic environmental parameters, current ionospheric delay data in the inner data region are used to estimate the ionospheric delay data in the outside region.

The estimation accuracy is evaluated at 12 points; four directions (north, south, east, and west) and three distances (5°, 10°, and 15°). The accuracy improvement by the SVM is compared with the NN. The one-month mean of the estimation error produced by the SVM is 0.33 TECU for the 5° region, 1.01 TECU for the 10° region, and 1.95 TECU for the 15° region. The improvement levels over the NN for the 5°, 10°, and 15° regions are 26.7%, 17.9%, and 5.3%, respectively. The error reduction by the SVM over NN is more significant at near points than at remote points.

Among the four directions, the error in the south region is the largest. The ionospheric delay and variation in the north region is usually smaller than the delay either in the east or west, but the extrapolation accuracy in the north region is even larger than in the east or west. A larger spatial gradient along the south-north direction over the east-west direction may explain this difference. This dependency on the ionospheric spatial gradient can be explained with inherent nature of extrapolation. A large gradient along the south-north direction implies a more sensitivity along the south-north direction data. The north point data is more sensitive to the southern part of input data than the western or eastern part of input data. Since the southern part of input data has a larger variation than other parts, its variation directly affects the north point estimate and increases the error.

Although artificial neural network is the most widely used machine learning algorithm for classification and regression problems, a SVM model is also powerful to prediction problem because of its generalization performance. Because a SVM is defined by a convex optimization problem, there are no local minima solutions. And SVM is based on structural risk minimization, it has excellent generalization performance. In case of our ionosphere extrapolation problem, the SVM demonstrates a better performance than the NN.

*Competing interests:* The authors declare that they have no conflict of interest.

*Acknowledgements.* This research was supported by the Space Core Technology Development Program funded by the Ministry of Science and Information and Communications Technology (ICT) (NRF-2016M1A3A3A02016943).

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
