# Peer review of "Extending the Coverage Area of Regional Ionosphere Maps Using a Support Vector Machine Algorithm"

_Annales Geophysicae, 2018_

## Referee Comment (RC1) · Anonymous Referee #1 · 18 Oct 2018

Extending the Coverage Area of Regional Ionosphere Maps Using a Support Vector Machine Algorithm The Ionospheric Delay Data (IDD) from a certain number of GNSS ground stations distributed inside an inner region, is exploited by the Support Vector Machine Algorithm (SVM) and Neural Networks (NN), to extrapolate IDD values in the outside region.

The results of SVM, NN, and BRDC (the Klobuchar model of the GPS navigation message), evaluated in terms of the RMS show unequivocally that SVM provides a higher prediction performance. The paper is well written and organized.

The possibility to extend, in real time, a regional coverage map of ionospheric de-

lay also where IDD are not available, through the extrapolation method proposed by authors, has an important scientific value because it represents an important info in Space Weather context. That's why I feel that this paper deserves to be published after some clarifications have been made and some incongruences resolved.

1) In the introduction (before the reference of Leandro and Santos) you write: "Ionospheric delay observations were used as the input parameters, and the TEC outside the coverage area was predicted". Therefore I would assume that the output is TEC, but looking at figure 4 on the y axis is written ionospheric delay. Therefore it would be better to eliminate this incongruence writing . . ... and the ionospheric delay outside the coverage area was predicted.

2) In section 2, Parameter modelling, you write "An extrapolated ionospheric delay, TECext, may be represented. . .." Why you label the ionospheric delay as TECext? Again, after Eq. (3), you talk about ionospheric delays but then in Eq. (4) you refer to observational parameters as TEC obs. It seems that you identified ionospheric delay with TEC. This sounds me strange because, to be meticulous, the ionospheric delay is proportional to the TEC therefore they are not exactly the same thing. To avoid this "incongruence" I suggest to replace in Eq. (4) TEC1 obs with ID1 obs, TEC2 obs with ID2 obs, TECN obs with IDN obs.

3) In section 3, subsection 3.1 again you write "y is the target that represent the true TEC in the extrapolation region". For the reasons written above I suggest to replace true TEC with true ionospheric delay.

4) In section 3, subsection 3.1, in Eq. (5 ), in the term WT the meaning of T is missing, T stands for? Moreover in Eq. (5) x should be replaced with xSVM to follow the same nomenclature adopted in the flow chart of figure 1 where you write xSVM = [xt xe xobs].

5) You write, at page 4 after figure 1, "Targets include the true TEC in the j-th extrapolation point", and looking at the flow chart of figure 1, I read ySVM = TECJext.. Therefore I ask you if it is not the case to replace in Eq. (6) and subsequently in the text, y with

ySVM .

6) At the end of subsection 3.1 you write "…using the interior point method", please provide a reference here.

7) At page 7 you write: "…the high ionospheric delay season is more appropriate when evaluating the extrapolation algorithm than the low ionospheric delay season". This sentence is not clear. Could you explain in other words this concept. What do you mean with "more appropriate ".

8) Looking at the caption of figure 4, I note some incongruences : a) you write "One year variation….(October 01, 2013 to October 30, 2014)" . This period is made by 13 months so it is a year + one month.

Probably it would be better to write the caption as: Ionospheric delay for the training period (01 October 2013 - 30 September 2014), and prediction period (01 - 31 October 2014). This is compatible with what you write before figure 4, i.e, "The training period is set to one year from October 1, 2013 to September 30, 2014."and "The prediction period is set to one month from October 1 to 31, 2014".

9) At the beginning of Section 5.1 you write "the variation of the TEC and the…". But given that the figure 5 shows on the y axis the ionospheric delay, again, I think it would be better to write: "ionospheric delay variations and the extrapolation results are analyzed for the data from October 28 2014, when…. "

10) In figure 5 the acronym BRDC is not defined, it should be defined in the caption.

11) After looking at figure 5, showing the ionospheric delay trend for the direction north (N5) and south (S5), I would have expected to see in figure 6 the extrapolation error trend still for the direction north and south, but strangely you provide the results for the east direction (E5). Why?

12) At pages 8 and 9 you provide the coordinates only for N5 and S5, then you mention the points E5, W5, N15, S15, E15, W15 without giving any info about their coordinates.

For completeness, I suggest to delete the info about the coordinates of N5 and S5, providing however a table where are reported all the coordinates of the key points involved in this analysis.

13) For completeness, in addition to figures 7 and 8, I suggest to insert also a new figure showing the results for 10° extrapolation regions.

14) With regard to Table 1, you write: "In the east and west points. . . . . . but the improvement increases as the distance of the extrapolation region increases". If I look at the errors for the direction east and west for SVM and NN, I note the following differences: 0.03, 0.20, and 0.13 (east direction), and 0.01, 0.01, and 0.17 (west direction) for 5°,10°, and 15° respectively. So, to be meticulous, it is not fully true what you say, because in the east direction 0.13 < 0.20 and in the west direction practically there is not difference between 5° and 10°. Therefore I suggest to delete the sentence "but the improvement increases as the distance of the extrapolation region increases".

References a) The reference Kim and kim (2014) is in the text, but it is missing in the list of reference

b) The reference Kim et al., 2014 is in the text, but it is missing in the list of reference

c) The reference Kim et al., 2014b is in the list of references, but it is missing in the text

d) Please replace in the list of references McKinella with McKinell

---

## Author Comment (AC1) · 25 Oct 2018

Thanks for your comments on this manuscript. The authors have incorporated all the comments in revised manuscript, which are very helpful to improve the manuscript. The terminology "TEC" has been replaced with "Ionospheric delay" to avoid possible confusion. One table has been added to describe the location of estimation points. One figure has been revised to match its previous figure. One figure has been added to discuss the estimation results in 10 degree. A detailed revision list and the revised manuscript are in the supplement file.

[Figure]

Please also note the supplement to this comment:
https://www.ann-geophys-discuss.net/angeo-2018-103/angeo-2018-103-AC1-
supplement.zip

---

## Referee Comment (RC2) · Anonymous Referee #1 · 26 Oct 2018

I read the revised version of the paper "Extending the Coverage Area of Regional Ionosphere Maps Using a Support Vector Machine Algorithm". The authors have been replied exhaustively to all the topics raised by the reviewer. My opinion is that the paper can be now published as it is on Annales Geophysicae.

---

## Author Comment (AC2) · 26 Oct 2018

Thank you again for your valuable comments and suggestions. They are very helpful for revising and improving our manuscript.

---

## Referee Comment (RC4) · Anonymous Referee #2 · 30 Oct 2018

The paper presents an extrapolative prediction capacity assessment of the Suport Vector Machine (SVM) and correlates it with predictions from Neural Networks (NN) and the Klobuchar Model. Results from the study suggest that the SVM gave better performance when compared to NN performance. The title is appropriate, and the abstract summarizes the intent and results of the research adequately.

I however make the following observations: 1. The performance of a NN model largely depends on the number of hidden layer neurons used. The authors indicate that they have used 80 hidden layer neurons based on previous studies. The previous study referenced does not give a convincing method to check overtraining of the networks.

[Figure]

Also the dataset is entirely different, and the NN architecture is also different. Using a different number of hidden layer neurons may give better results, perhaps better than the SVM method. I therefore suggest that the authors device a system to check performance of the networks (especially on extrapolation datasets), if not, the networks may even over-fit the training data and so perform poorly on extrapolation data. The authors may also choose to indicate/explain in the manuscript that the observation they report is not generalized (but limited to the case of their NN training) because a carefully done NN may give better results, even than the SVM does.

2. There is also information which appears missing in the manuscript. Inputs for the models do not include station locations? How do the models predict different values for different locations? The spatial structure (with station locations) is pre-fixed in the models? How do you query the models for data of, let's say, 10 degrees from the center of your circle? I wonder what applications there are for this method if the spatial structure for the models is pre-fixed.

3. Although the authors have used data for South Korea, they do not indicate the implication of this limitation anywhere on the manuscript. Given the spatial variability of the ionosphere, extrapolation schemes for a given region will perform differently for different regional models. For instance, whether the ionospheric ionization should be greater or otherwise in the outer regions is something too arbitrary to decide based on the inner data. And if the outer data will always be required to train the relationship, then the application I see of this work is defeated.

4. Page 2, lines 32-33: It is not clear why two solar activity indicators (F10.7 and SSN) are repeated. Also, how does the method in this work take care of the time lag (up to several hours/days) for geomagnetic storm effects to be observed in the ionosphere?

5. Page 1, lines 37-38: "Kim and Kim (2016) additionally used ionospheric delays in the inner ionospheric coverage area." It is not clear what this sentence means, and why it is necessary to include it here.

6. Page 3, line 16: "In the above equation..." should read " In equation 7...".

7. Consider using "ionospheric map/model" in places of "ionosphere map/model" throughout the manuscript.

8. Page 7, line 1: Authors should clarify what previous one-epoch values are referred. What is the interval between successive epochs? Is the interval between successive epochs sufficiently small for previous one-epochs to be safely used? And what happens if there may be no data for previous one, two, three. . .. epochs?

9. The authors cite SVM applications to other fields but not a citation on previous ionospheric applications. There have been previous studies on the use of SVM for Ionospheric research. E.g.: https://agupubs.onlinelibrary.wiley.com/doi/full/10.1029/2010RS004393 https://agupubs.onlinelibrary.wiley.com/doi/full/10.1029/2010RS004633 https://www.ann-geophys.net/31/173/2013/angeo-31-173-2013.pdf

---

## Referee Comment (RC5) · Anonymous Referee #3 · 5 Nov 2018

REVIEWER_1 COMMENTS

This paper try to explain extending the coverage area TEC data enterpolation using SVM algrorithm. Therfore this paper is important for building a system to obtain more precise TEC value and its application. I will suggest some revision to be publihed in annales geop. 1- In page 7 line 2 you say that correlation coefficient between the two adjacent data for F10.7 cm, Kp and SSN is 0.93, 0.863, 0.852. I think that correlation analysis should be between F10.7 cm-TEC, Kp-TEC, SSN-TEC. According to these results, you will do weighting and this weighthing affects your results. The other thing is that F10.7 cm is only one value for one day, there-

fore its affect can not be monitored effectively in a day. You can obtain Kp value from https://omniweb.gsfc.nasa.gov/form/dx1.html with one hour resolution. You should also take into account DsT index which give information about geomagnetic activity of ionosphere. You can also more precise results by adding this DsT indexs 2- In this paper, I can not see any discussion, therefore your results can not be confirmed. Please investigate other studies and compare your result. 3- Conclusion also should be explained detaily. You mention about differences but you didnt any comment on this. Please explain these differences.

---

## Author Comment (AC3) · 12 Nov 2018

Thanks for your comments on this manuscript. The authors have incorporated all the comments in revised manuscript. Two figures have been added to describe the parameter selection process. Several paragraphs have been added to clarify the estimation procedure and to explain some detailed aspects. The manuscript has been revised from the first revision posted on October 25 (Respond to Referee #1). A detailed revision list and the revised manuscript are in the supplement file.

Please also note the supplement to this comment:

[Figure]

https://www.ann-geophys-discuss.net/angeo-2018-103/angeo-2018-103-AC3-supplement.zip

---

## Author Comment (AC4) · 19 Nov 2018

Thanks for your comments on this manuscript. The authors have incorporated all the comments in revised manuscript. Series of experiments has been performed after replacing Kp with Dst as the referee suggested. Several paragraphs have been added to enhance the analysis of the results. The conclusion has been revised for more discussion on the results. The manuscript has been revised from the second revision posted on November 12 (Respond to Referee #2). A detailed revision list and the revised manuscript are in the supplement file.

[Figure]

Please also note the supplement to this comment:
https://www.ann-geophys-discuss.net/angeo-2018-103/angeo-2018-103-AC4-supplement.zip
* * *
Interactive
comment

---

## Author Comment (AC5) · 26 Nov 2018

Thank you again for your valuable comments and suggestions. They are very helpful for revising and improving our manuscript.

---

## Author Response (AR1)

**REVISION LIST (Topical Editor)**

Title:Extending the Coverage Area of Regional Ionosphere Maps Using a Support Vector
Machine AlgorithmAuthors:Mingyu Kim and Jeongrae KimDate:December 18, 2018

Dear topical editor,

Thanks for your comments on this manuscript. The authors have incorporated all the comments in revised manuscript. The revised or new sentences are colored in red in the revised manuscript. The manuscript has been revised from the third revision posted on November 19 (Response to Referee #3), which incorporated all comments from three referees. Previous responses to referees #1, #2, and #3 are attached after this response letter.

**< Editor >**

**1)** Page 2, paragraph 25:

The TEC variation is correlated with the diurnal and seasonal time variation, and the ionospheric delay above the locations involved in the study reaches its maximum around 14 hours local time (LT) and its minimum around 2 LT. Also, the TEC is higher in spring and autumn, and lower in summer and winter (here you mean the same local time?).

Thanks for the correction. The sentences have been updated. The TEC variation in the second sentence implies a daily mean TEC variation. "daily mean" has been added to the TEC.

<Sec.2, p.2>

2) I agree with the referee's comment that using both F10.7 and SSN is not necessary, or you need to argue the necessity/importance of using both indices.

Two additional set of estimations have been performed; (a) with F10.7 only and (b) with SSN only. The authors had already performed these estimations for answering referee #2's comment. This time more comprehensive runs have been performed. Each NN/SVM estimator has been configured as an optimal structure after series of parameter tuning.

The revised results still show that the dual parameter case (F10.7+SSN) outperforms the single

cases (F10.7 only or SSN only). The variation of F10.7 and SSN are very similar, but they are not the same quantity. The variation of two parameters are somewhat different and their complimentary aspect may improve the estimation performance. One of the advantages of machine learning is automatic determination of parameter weightings, and a high correlation between input parameters is not a significant problem as conventional linear regressions. Another aspect is the limitation of the data length. The results are based on one-month data. If we process multiple years of long data under different space environment, the results might be changed.

The estimation results of two single parameter cases, with F10.7 only and with SSN only, are added at the end of Section 5. Full description of the results are at the end of this letter.

<Sec.5, pp.14-15>

3) As for using *Kp* and *Dst* indices, there it is necessary to take into account different sources of ionospheric disturbances CMEs and CIR/CH HSSS. CMEs induce non-recurrent storms, while recurrent storms are driven by high-speed solar wind and reappearing with about 27-day periodicity, when the same coronal hole (CH) is facing the Earth. During this kind of disturbance the *Dst* index remains smaller, but because fast streams with southward IMF component may last much longer, the CIR/HSSS related storms have a longer duration, and the cumulative effects of these storms could be more severe than the effects of CME-related storms with significant decrease in *Dst*. (Buresova, et al., 2014). So in the case of coronal holes you need to monitor polar activity (e.g. *Kp*, *AE*, unfortunately, is not available in real time)

Thanks for your information. The difference between Kp and Dst may explain the estimation results using Dst (instead of Kp), which was presented in the letter to Referee #3's comments. Using Kp was better than Dst in the estimation results. Much longer data span, e.g. multiple years, may be required to analyze the Kp and Dst parameter effects. It can be a good for further research. Sentences are added to discuss this aspect.

<Sec.2, pp.2-3> <Sec.4, pp.10> <Ref., pp.16>

**< Supplement for comment #1 >**

**(a) Parameter optimizations for two single parameter cases (F10.7 or SSN)**

Estimation accuracy by NN or SVM can be affected by the design of estimator structures;

parameter value C or number of layers, etc. Optimal estimator structure is changing with the selection of input parameters. Before comparing the single parameter (F10.7 only or SNN only) results with the dual parameter (F10.7 and SNN) results, series of parameter optimizations have been performed for determining an optimal estimator for NN or SVM.

Figure 1. SVM test errors of different C values at S5; F10.7 only (left) and SSN only (right)

Figure 1 shows the optimization results of SVM models with F10.7 or SSN. The optimal C values are computed as the lowest mean RMS errors on the 5° extrapolation points. For the both single parameter cases, an optimal C value is determined to 10000.

Figure 2. NN test errors of different number of neurons at S5; F10.7 only (left) and SSN only (right)

The optimization results of NN model for each F10.7 and SSN are shown in Fig. 2. The optimal numbers of hidden neurons are 55 for the F10.7 case and 45 for the SSN case.

---

## Editor Decision (ED1)

Dear Authors,

I am coming back to you on the status of your paper. Thank you for your response to the referees' comments. All three referees found your paper as interesting and contributive to the ionospheric research. The argumentation in the manuscript is given in an easy way, is clear and is easy to follow. There was also pointed out in the reports that the work you are presenting in the manuscript is important step towards a more precise representation of the state of ionization of the ionosphere. However, referees had some important objections to the present version of the manuscript, and this is a reason why I suggest a minor revision. Please, consider carefully and discuss in the revised version of the manuscript all comments of the referees.

From my side, I would like to draw your attention to:

i)      Ppage 2, paragraph 25:

The TEC variation is correlated with the diurnal and seasonal time variation, and the ionospheric delay above the locations involved in the study reaches its maximum around 14 hours local time (LT) and its minimum around 2 LT. Also, the TEC is higher in spring and autumn, and lower in summer and winter (here you mean the same local time?).

ii)     I agree with the referee's comment that using both F10.7 and SSN is not necessary, or you need to argue the necessity/importance of using both indices.

iii)    As for using *Kp* and *Dst* indices, there it is necessary to take into account different sources of ionospheric disturbances CMEs and CIR/CH HSSS. CMEs induce non-recurrent storms, while recurrent storms are driven by high-speed solar wind and reappearing with about 27-day periodicity, when the same coronal hole (CH) is facing the Earth. During this kind of disturbance the *Dst* index remains smaller, but because fast streams with southward IMF component may last much longer, the CIR/HSSS related storms have a longer duration, and the cumulative effects of these storms could be more severe than the effects of CME-related storms with significant decrease in *Dst*. (Buresova, et al., 2014). So in the case of coronal holes you need to monitor polar activity (e.g. *Kp, AE* , unfortunately, is not available in real time)

If you are prepared to undertake the improvements required, please submit the revised manuscript as well as a list of changes or a rebuttal against each point, which is being raised when you submit the revised manuscript.

Kindest regards

Yours sincerely

D. Buresova

---

## Author Response (AR2)

**REVISION LIST (Topical Editor)**

Title: Extending the Coverage Area of Regional Ionosphere Maps Using a Support Vector Machine Algorithm

Authors: Mingyu Kim and Jeongrae Kim

Date: January 09, 2019

Dear topical editor

Thanks for your comments on this manuscript. The authors have incorporated all the comments in revised manuscript. The revised or new sentences are colored in red in the revised manuscript. The manuscript has been revised from the minor revision posted on December 21 (Response to Editor).

**< Topical Editor >**

**1)** Page 1, Paragraph 15:

What indices do you mean in the last sentence of the paragraph "...hourly/daily indices...". It should be clarified.

It two variables; observation hour and day number. The sentence has been revised.

<Sec.1, p.1>

**2)** Page 2, paragraph 25:

Please, give some references to the results published by you or to other scientific papers for the statement "*The ionospheric variation is correlated with the diurnal and seasonal time variation, and the ionospheric delay above the locations involved in the study reaches its maximum around 14 hours local time (LT) and its minimum around 2 LT. Also, the daily mean ionospheric delay is higher in spring and autumn, and lower in summer and winter.*"

We have added two references. (Wu et al. 2012, Mansoori et al. 2015).

<Sec. 2, p.2>, <Ref., p.17>

**3)** Page 2, paragraph 40:

The sentence "However, Dst response performance depends on ionosphere storm types" is not correct. It depends of the ionospheric storm driver.

Thanks for your tip. "Types" has been changed to "driver".

<Sec.2, p.3>

**4)** Page 3, the second sentence from the top:

*"After performing some numerical experiments with Dst, Kp has been selected for the parameter."*

and pages 9-10, paragraph 40:

*"Series of experiments had been performed by using Dst instead of Kp. In certain instances Dst reflects ionospheric variation better than Kp does. A correlation analysis between Dst or Kp with TEC showed that Dst yields a slightly higher correlation values than Kp. Therefore, we had performed another estimation process after replacing Kp with Dst. Our preliminary results showed that Dst is not better than Kp for our estimation algorithm. The results may be different for another data period when CME-driven ionosphere storm occurs. One month of data period tested in this research may not be sufficient for determining optimal parameter. Comprehensive analysis with a longer data period, e.g. multiple years, will be helpful."*

When reading what was said in both page 3 and pages 9-10, there is not clear, what index you are finally using in your computation. Please, give clear information for readers. The statement that "…Dst reflects ionospheric variations better than Kp does" is also not correct. In this case wording "…storm-time variations.." is more clear. In general, the Kp index is usually more suitable, as it reflects both CME- and CIR/CH HSS-related ionospheric disturbances (or combination of both Kp and Dst). On the other hand, as you have mentioned, one-month data analysis is really not enough to get a clear indication of a reliability of the indices. Also geomagnetic activity during October 2014 was minor–to-moderate, what is not an excellent choice when evaluating correlation.

Kp was used for the final computation. The two paragraphs in page 3 and 9/10 have been fully revised to clarify the parameter selection. The shortcoming of Dst has been stated and the test results between Kp and Dst have been briefly included in the paragraph.

<Sec.2, p.3> <Sec. 4, pp. 9-10>

**5)** Page 3, paragraph 40:

What do you mean using the term *"inner ionosphere"*? The term should be clarified when it was used first.

The inner ionosphere represents a geographical area where ionospheric delay information or observations are available. We have added the definition of the inner and outer areas and replaced "inner ionosphere map data" with "ionosphere input data" for clarification.

<Sec.2, p.2, p.3, p.7, p.8, p.12>

---

## Editor Decision (ED2)

Dear Authors,

Thank you for considering comments of the referees and for sending us the improved manuscript. I have read it carefully and found some points to be clarified before the manuscript will be recommended for publication.

*Page 1, paragraph 15*:

What indices do you mean in the last sentence of the paragraph "…hourly/daily indices…". It should be clarified.

*Page 2, paragraph 25:*

Please, give some references to the results published by you or to other scientific papers for the statement "*The ionospheric variation is correlated with the diurnal and seasonal time variation, and the ionospheric delay above the locations involved in the study reaches its maximum around 14 hours local time (LT) and its minimum around 2 LT. Also, the daily mean ionospheric delay is higher in spring and autumn, and lower in 25 summer and winter.*

Page 2, paragraph 40:

The sentence *"However, Dst response performance depends on ionosphere storm types"* is not correct. It depends of the ionospheric storm driver.

*Page 3, the second sentence from the top:*

*"After performing some numerical experiments with Dst, Kp has been selected for the parameter."*

*and pages 9-10, paragraph 40:*

*"Series of experiments had been performed by using Dst instead of Kp. In certain instances Dst reflects ionospheric variation better than Kp does. A correlation analysis between Dst or Kp with TEC showed that Dst yields a slightly higher correlation values than Kp. Therefore, we had performed another estimation process after replacing Kp with Dst. Our preliminary results showed that Dst is not better than Kp for our estimation algorithm. The results may be different for another data period when CME-driven ionosphere storm occurs. One month of data period tested in this research may not be sufficient for determining optimal parameter. Comprehensive analysis with a longer data period, e.g. multiple years, will be helpful."*

When reading what was said in both page 3 and pages 9-10, there is not clear, what index you are finally using in your computation. Please, give clear information for readers.

The statement that "…Dst reflects ionospheric variations better than Kp does" is also not correct. In this case wording "…storm-time variations.." is more clear. In general, the Kp index is usually more suitable, as it reflects both CME- and CIR/CH HSS-related ionospheric disturbances (or combination of both Kp and Dst). On the other hand, as you have mentioned, one-month data analysis is really not enough to get a clear indication of a reliability of the indices. Also geomagnetic activity during October 2014 was minor–to-moderate, what is not an excellent choice when evaluating correlation.

*Page 3, paragraph 40*:

What do you mean using the term "*inner ionosphere*"? The term should be clarified when it was used first.